# Trioleyl Pyridinium, a Cationic Transfection Agent for the Lipofection of Therapeutic Oligonucleotides into Mammalian Cells

**DOI:** 10.3390/pharmaceutics15020420

**Published:** 2023-01-27

**Authors:** Ana Delgado, Rosa Griera, Núria Llor, Ester López-Aguilar, Maria Antònia Busquets, Véronique Noé, Carlos J. Ciudad

**Affiliations:** 1Department of Biochemistry and Physiology, School of Pharmacy and Food Sciences, University of Barcelona, 08028 Barcelona, Spain; 2Department of Pharmacology, Toxicology and Therapeutic Chemistry, School of Pharmacy and Food Sciences, University of Barcelona, 08028 Barcelona, Spain; 3Department of Pharmacy and Pharmaceutical Technology and Physical Chemistry, School of Pharmacy and Food Sciences, University of Barcelona, 08028 Barcelona, Spain; 4Nanoscience and Nanotechnology Institute, IN2UB, University of Barcelona, 08028 Barcelona, Spain

**Keywords:** therapeutic oligonucleotide, PPRH, hairpin, delivery, transfection agent, mammalian cell, cancer, cell viability, apoptosis, stable transfection

## Abstract

Background: One of the most significant limitations that therapeutic oligonucleotides present is the development of specific and efficient delivery vectors for the internalization of nucleic acids into cells. Therefore, there is a need for the development of new transfection agents that ensure a proper and efficient delivery into mammalian cells. Methods: We describe the synthesis of 1,3,5-tris[(4-oelyl-1-pyridinio)methyl]benzene tribromide (TROPY) and proceeded to the validation of its binding capacity toward oligonucleotides, the internalization of DNA into the cells, the effect on cell viability, apoptosis, and its capability to transfect plasmid DNA. Results: The synthesis and chemical characterization of TROPY, which can bind DNA and transfect oligonucleotides into mammalian cells through clathrin and caveolin-mediated endocytosis, are described. Using a PPRH against the antiapoptotic *survivin* gene as a model, we validated that the complex TROPY–PPRH decreased cell viability in human cancer cells, increased apoptosis, and reduced survivin mRNA and protein levels. TROPY was also able to stably transfect plasmid DNA, as demonstrated by the formation of viable colonies upon the transfection of a *dhfr* minigene into *dhfr*-negative cells and the subsequent metabolic selection. Conclusions: TROPY is an efficient transfecting agent that allows the delivery of therapeutic oligonucleotides, such as PPRHs and plasmid DNA, inside mammalian cells.

## 1. Introduction

Nucleic acids could be a promising alternative for the treatment of genetic diseases such as cancer [1], cardiovascular, neurological [2,3], or hematological [4,5] disorders since oligonucleotides can modulate gene expression. The Food and Drug Administration (FDA) approved the use of therapeutic oligonucleotides [6,7] including antisense oligonucleotides [8], small interfering RNAs (siRNA), and aptamers [9] as nucleic acids tools [10]. In our laboratory, we developed a new kind of therapeutic oligonucleotide named PolyPurine Reverse Hoogsteen (PPRH) hairpins [11,12]. PPRHs are nonmodified DNA molecules, formed by two specular arms of more than twenty nucleotides of polypurines linked by a thymidine loop. The two strands run in an antiparallel fashion and are bound by intramolecular Hoogsteen bonds between adenines (A-A) and guanines (G-G), thus adopting a hairpin disposition. The polypurine sequence is designed to hybridize, in a sequence-specific manner, the polypyrimidine strand of a dsDNA sequence in the desired gene by Watson and Crick bonds [13]. The formation of the triplex between the two mirror repeat domains of the PPRH and the polypyrimidine sequence displaces the fourth strand of the target dsDNA. This causes transcription inhibition of the targeted gene, thus modulating its expression. The capacity of PPRHs to silence the expression of specific genes has been validated both in vitro [11,14] and in vivo [15] in our laboratory using cancer cell lines, directing the oligonucleotides to specific gene targets that are involved in the development of cancer. PPRHs are targeted toward polypyrimidine sequences found in promoters, intronic, or exon regions. These targets can be found in practically all genes. PPRHs present a series of advantages in comparison to the other therapeutic oligonucleotides mentioned above. PPRH molecules are more stable than siRNAs because their hairpin structure is composed of deoxynucleotides instead of ribonucleotides [16]. Furthermore, PPRHs bind dsDNA with higher affinity compared to triplex-forming oligonucleotides (TFOs) [13]. At low concentrations, they are more effective than ASOs [17]. Finally, PPRHs do not show immunogenicity [18], hepatotoxicity, or nephrotoxicity [19]. In this work, we used a PPRH directed toward the promoter sequence of *survivin* (BIRC5), an antiapoptotic gene overexpressed in many cancers, which produces a high degree of cytotoxicity due to an increase in apoptosis.

One of the most significant limitations of therapeutic oligonucleotides is the development of specific and efficient delivery vectors that can guarantee the internalization of the nucleic acids into cells [20]. Usually, transfection methods are classified into viral, physical, and chemically synthesized molecules [21,22,23]. Viral vectors, such as retrovirus or lentivirus, have a high delivery efficiency. However, they present a size limitation for the transgene they transmit [24]. Furthermore, viral agents can cause alterations in the DNA or generate immunogenic responses in the organism [25]. Other alternatives for the transfection of therapeutic oligonucleotides are physical systems based on cell membrane deformation. On one hand, these methods are safer than viral agents; on the other hand, they present lower transfection efficiency, and in some cases, they compromise cell viability [26,27]. Additionally, it would be difficult to scale them for the treatment of internal organs. Finally, chemical compounds are easier to produce and are susceptible to modification, increasing the possibility of directing the delivery to specific tissues, and new nanotherapeutic systems have been recently developed for nucleic acid delivery in mammalian cells [21,22,28]. Previous experiments of gene silencing were performed in our laboratory using a commercially available cationic liposome *N*-[1-(2,3-dioleoyloxy)propyl]-*N*,*N*,*N*-trimethylammonium methylsulfate (DOTAP) for the delivery of PPRHs [29,30]. However, in some cell lines such as neuroblastoma (SH-SY5Y), which are hard to transfect, the transfection was ineffective [31].

In recent years, the research of the Organic Chemistry section of our group has been oriented toward the synthesis of compounds that could be used as nonviral delivery systems. Recently, we validated the effectiveness of 1,3-bis[(4-oleyl-1-pyridinio)methyl]benzene dibromide (DOPY) [31] to deliver PPRHs in different cancer cell lines, including SH-SY5Y. In this work, we describe the synthesis of 1,3,5-tris[(4-oelyl-1-pyridinio)methyl]benzene tribromide (TROPY) and validate it as a nonviral vehicle that can transport and deliver therapeutic oligonucleotides PPRHs in cancer cells.

## 2. Materials and Methods

### 2.1. Chemicals and Instrumentation

1,3,5-Tris(bromomethyl)benzene, acetone, acetonitrile and deuterochloroform (CDCl_3_) were purchased from Sigma-Aldrich. All chemicals were of analytical grade and used directly without any further modification. 4-Oleylpyridine was prepared according to the procedure in the literature [31].

A hot plate magnetic stirrer with an aluminum heating block was used to heat the reaction. Evaporation of the solvent was accomplished with a rotary evaporator. The reaction course was routinely monitored by thin-layer chromatography on SiO_2_ (silica gel 60 F254), and the spots were located by either a UV light or a 1% KMnO_4_ solution.

NMR spectra were recorded at 400 MHz (^1^H) and 100.6 MHz (^13^C) on a Bruker Avance Neo 400 from Serveis Científics i Tecnològics de la Universitat de Barcelona (CCiT-UB); chemical shifts are reported in δ values, in parts per million (ppm) relative to Me_4_Si (0 ppm), or relative to residual chloroform (7.26 ppm, 77.00 ppm) as an internal standard. Data are reported in the following manner: chemical shift, multiplicity, coupling constant (*J*) in hertz (Hz), and integrated intensity.

High-resolution mass spectra (HMRS) were performed by CCiT-UB using an electrospray (ESI) ionization source and a TOF analyzer.

### 2.2. Preparation of TROPY

1,3,5-Tris(bromomethyl)benzene (35 mg, 0.096 mmol, 97%) was added to a stirring solution of 4-oleyloxypyridine (100 mg, 0.289 mmol) in acetone (5 mL), and the mixture was heated at reflux for 48 h. After cooling to room temperature, the solvent was evaporated under reduced pressure, and the crude was suspended in acetonitrile (5 mL) and stirred for 10 min. The resulting residue was separated by decantation, affording 1,3,5-tris[(4-oelyl-1-pyridinio)methyl]benzene tribromide (TROPY) (107 mg, 80%, MW 1394.88 g/mol) as a sticky off-white solid. The structure of TROPY was characterized and confirmed by ^1^H NMR and ^13^C NMR spectroscopy, and HRMS.

^1^ H NMR (400 MHz, CDCl_3_) δ 9.95 (d, *J* = 6.7 Hz, 6H), 9.03 (br s, 3H), 7.34 (d, *J* = 6.7 Hz, 6H), 5.71 (br s, 6H), 5.36-5.33 (m, 6H), 4.19 (t, *J* = 6.5 Hz, 6H), 2.03–1.98 (m, 12H), 1.84 (quint, *J* = 7 Hz, 6H), 1.47–1.39 (m, 6H), 1.38–1.21 (m, 60H), 0.87 (t, *J* = 6.5 Hz, 9H).

^13^C NMR (100.6 MHz, CDCl_3_) δ: 170.4, 146.9, 135.7, 132.5, 129.9, 129.6, 114.2, 71.4, 61.1, 31.8, 29.7, 29.6, 29.4, 29.3, 29.2, 29.1, 29.0, 28.3, 27.1, 27.0, 25.5, 22.6, and 14.0.

HRMS (ESI-TOF) *m*/*z*: calculated for [C7_8_H_126_N_3_O_3_] ^+3^ 384.3261; found 384.3266.

### 2.3. Design and Usage of PPRHs

The search for polypurine sequences to design PPRHs from was performed with the triplex-forming oligonucleotide target sequence search software (http://utw10685.utweb.utexas.edu/tfo/, accessed on 22 December 2022, MD Anderson cancer center, The University of Texas) [32]. The PPRH (HpsPr-C) against the *survivin* gene, which was directed to the promoter sequence, was used for gene silencing experiments and was previously validated in our laboratory [15,19]. The design of the hairpin consisted of two antiparallel mirror repeats of polypurine sequences bound by intramolecular reverse Hoogsteen bonds and linked by a thymidine loop (4–5 Ts). The same PPRH against *survivin* was also synthesized with a fluorescence label (FAM-HpsPr-C) in its 5′-end. The negative control was an unspecific scrambled hairpin (HpScr9) that could not form a triplex with the target DNA.

Hairpins used in the study were synthesized as nonmodified oligodeoxynucleotides by Merck (Haverhill, UK). They were resuspended in a sterile Tris–EDTA buffer (1 mM EDTA and 10 mM Tris, pH 8.0) (Merck, Madrid, Spain) and stored at −20 °C. The hairpin sequences are shown in Table 1.

The PPRHs designed using the TFO searching tool are shown in Figure 1. The two sequences run in an antiparallel fashion bound by Hoogsteen bonds and linked by a thymidine loop.

### 2.4. Agarose Gel Retardation Assays

The binding capacity of TROPY toward DNA was analyzed in reaction mixtures containing 100 ng of FAM-HpScr9 and increasing amounts of TROPY in a final volume of 10 µL. After 20 min at room temperature, the samples were electrophoresed in 0.8% agarose gels in TAE buffer 1X. Gels were visualized on Gel Doc^TM^ EZ (Bio-Rad Laboratories, Inc., Barcelona, Spain).

### 2.5. Cell Culture

The different cell lines used for gene silencing experiments were SH-SY5Y, PC-3, MDA-MB-453-WT, HepG2, CHO, and Vero-E6. Cell lines were obtained from the Cell Bank resources of the University of Barcelona. Cells were grown in Ham’s F12 medium supplemented with 10% fetal bovine serum or in RPMI medium supplemented with 7% dialyzed fetal bovine serum for the experiments of plasmid transfection (all from GIBCO, Invitrogen, Barcelona, Spain). Cells were incubated at 37 °C in a humidified 5% CO_2_ atmosphere. Subculture was performed with Trypsin 0.05% (Merck, Madrid, Spain).

### 2.6. Transfection of DNA

Cells were plated in 6-well dishes in 900 µL of F12 medium. After 24 h, cells were transfected with a mix containing different amounts of TROPY, 100 nM of the PPRHs, and serum-free medium up to 100 μL. After 20 min of incubation, the mix was added to the cells to achieve a final volume of 1 mL in full medium (Ham’s F12 medium). In the case of plasmid DNA, the transfection was performed using pDHC1P as a vector for dihydrofolate reductase (DHFR) transfected into *dhfr*-negative DG44 cells [33] and metabolically selected in a selected in a selective medium lacking hypoxanthine and thymidine(RPMI) until positive cells developed, usually two weeks. Colonies were allowed to grow for an additional week, then fixed with 2% formaldehyde, stained with crystal violet, and counted.

### 2.7. DLS and Z-Potencial

The size of TROPY/PPRH nanoparticles was measured by dynamic light scattering (DLS) with a Zetasizer Nano (Malvern, UK) at a fixed angle of 90°. The incubation of the lipofection agent TROPY plus the PPRH against *survivin* (100 nM) was carried out by reproducing the same conditions used in the transfection. The mixture was prepared in 100 µL of water and was diluted up to 1 mL just before the measurements.

The charge of TROPY in the presence of the oligonucleotide was determined by measuring the ζ-potential using Doppler microelectrophoresis using a Zetasizer Nano (Malvern, UK). The final volume of the sample was 1.2 mL.

### 2.8. Cellular Uptake

The internalization of the nanoparticles was evaluated by flow cytometry. PC-3, SH-SY5Y, and MDA-MB-453-WT (90,000) cells were plated in 6-well dishes in F12 medium 24 h before the transfection. Then, transfection was carried out using FAM-Src-9 PPRH (100 nM) in combination with the corresponding amount of TROPY. After 24 h of incubation, cells were trypsinized and collected in cold PBS, centrifuged at 800× *g* at 4 °C for 5 min, and washed once in PBS. The pellet was resuspended in 400 μL PBS, and propidium iodide was added to a final concentration of 5 µg/mL (Merck, Madrid, Spain). Flow cytometry analyses were performed at the CCiT-UB in a Gallios flow cytometer (Beckman Coulter, Inc., Barcelona, Spain).

Additionally, fluorescence microscopy cell images were acquired using a ZOE Fluorescent Cell Imager (Bio-Rad Laboratories, Inc., Barcelona, Spain) just before flow cytometer analyses.

To study the internalization mechanism of TROPY, PC-3 cells (120,000) were plated in 6-well dishes in the F12 medium. After 24 h, the cells were preincubated with 75 μM of the clathrin-dependent endocytosis inhibitor, Dynasore [34], 185 μM of the caveolin-mediated endocytosis inhibitor, Genistein [35], or 33 μM of the micropinocytosis inhibitor, 5-(N-ethyl-N-isopropyl) amiloride (EIPA) [36], all from Merck, Madrid, Spain, for 60 min at 37 °C. Then, transfection mixes containing FAM-HP-Scr9 were added to cells for 3.5 h and processed for flow cytometry analyses, as described above in this section.

### 2.9. Confocal Analyses

The internalizations of TROPY–PPRH complexes were analyzed using confocal microscopy in PC3 cells. In this case, 60,000 cells were plated and incubated with 1 µg/µL of TROPY plus 100 nM of the scrambled PPRH. After 24 h of incubation, cells were cooled at 4 °C for 5 min and washed twice with PBS for 5 min. PC-3 cells were incubated for 30 min with two fluorescent dyes, *WGA 555* (to label nuclei in red) and *Hoechst* (to label membranes in blue). Finally, cells were washed with PBS at room temperature for 5 min and visualized using confocal microscopy at the CCiT-UB.

### 2.10. MTT Assays

Cells (10,000) were plated in 6-well dishes in 1 mL F12 medium and incubated with the oligonucleotide plus the liposome. Five days after transfection, cell viability was determined by adding to each well 0.63 mM of 3-(4,5- dimetilthyazol-2-yl)-2,5-dipheniltetrazolium bromide and 100 μM sodium succinate (both from Merck, Madrid, Spain). After 2.5 h at 37 °C of incubation, the culture medium was removed and the lysis solution was added (0.57% of acetic acid and 10% of sodium dodecyl sulfate (SDS) in dimethyl sulfoxide) (Merck, Madrid, Spain). Absorbance was measured at 570 nm in a Varioskan Lux, Thermo Scientific, Barcelona, Spain. Cell viability was expressed as the percentage of cell survival, relative to the controls.

### 2.11. Analysis of Survivin Expression

#### 2.11.1. Survivin mRNA Levels

Total RNA from control or PPRH-transfected cells for 24 h and 48 h was extracted using Trizol Reagent (Life Technologies, Madrid, Spain), following the instructions of the manufacturer. Complementary DNA was synthesized in a 20 µL reaction mixture from 1 µg of total RNA, 0.5 mM of each deoxyribonucleotide triphosphate (dNTP, Epicentre, Madison, USA), 250 ng of random hexamers (Roche, Barcelona, Spain), 10 mM dithiothreitol, 200 units of a Moloney murine leukemia virus reverse transcriptase (RT), 20 units of RNase inhibitor, and 4 µL of buffer (5×) (all three from Lucigen, Middleton, WI, USA). The reaction was incubated at 42 °C for 1 h.

The BIRC5 mRNA TaqMan probe (Hs04194392_s1; ThermoFisher Scientific, Madrid, Spain) was used to determine survivin mRNA levels and the Cyclophilin (PP1A) mRNA TaqMan probe (Hs04194521_s1, ThermoFisher Scientific, Madrid, Spain) was used as the endogenous control. The reaction was conducted in 20 μL containing 1xTaqMan Universal PCR Mastermix (Applied Biosystems, Madrid, Spain), 0.5xTaqMan probe, and 3 μL of cDNA. PCR cycling conditions were 10 min denaturation at 95 °C, followed by 40 cycles of 15 s at 95 °C, and 1 min at 60 °C using a QuantStudio 3 Real-Time PCR System (Applied Biosystems, Barcelona, Spain). The quantification was performed using the ΔΔCt method, where Ct is the threshold cycle that corresponds to the cycle when the amount of amplified mRNA reaches the fluorescence threshold.

#### 2.11.2. Survivin Protein Levels

Protein extracts from PC-3 (60,000) were obtained 24 h after transfection with TROPY–PPRH complexes. Cells were collected in 100 µL of Lysis buffer (0.5 M NaCl, 1.5 mM MgCl_2_, 1 mM EDTA, 10% glycerol, 1% Triton X-100, and 50 mM HEPES, pH 7.2), supplemented with a protease inhibitor mixture (P8340) (all from Merck, Madrid, Spain). Protein extracts were kept on ice for 1 h with vortexing every 15 min. Cell debris was removed by centrifugation (16,300× *g* for 10 min).

Protein concentration was determined using a Bio-Rad protein assay based on the Bradford method and using bovine serum albumin as a standard. Whole-protein extracts (100 µg) were electrophoresed in a 15% SDS-polyacrylamide gel and transferred to Immobilon-P polyvinylidene difluoride membranes (Merck, Madrid, Spain) using a semidry electroblotting system. Blocking was performed using a 5% skimmed milk solution. Membranes were probed with a primary antibody against survivin (5 µg/mL; AF886, Bio-Techne R&D Systems, S.L.U., Madrid, Spain), or against GAPDH (1:250 dilution; sc-47724, Santa Cruz Biotechnology, Heidelberg, Germany). Secondary horseradish peroxidase-conjugated antibodies were anti-rabbit (1:2000 dilution; P0399, Dako, Denmark) for survivin and anti-mouse (1:2500 dilution; sc-516102, Santa Cruz Biotechnology, Heidelberg, Germany) for GAPDH detection. Chemiluminescence was detected with the ImageQuant LAS 4000 mini (GE Healthcare, Barcelona, Spain). Quantification was performed using ImageQuant 5.2 software.

### 2.12. Apoptosis Assays

Cells (90,000) were incubated for 48 h with the combination of PPRH plus the transfectant agent, trypsinized, and collected in PBS. Then, cells were centrifuged at 1200× *g* for 5 min. The pellet was resuspended in 100 μL of Binding Buffer 1X. Samples were incubated with the APC Annexin V Detection Kit (Invitrogen, Thermo Scientific, Barcelona, Spain) for 15 min at room temperature. After 5 min of centrifugation, the pellet was washed in 1 mL of binding buffer. Finally, samples were centrifugated for 5 min and resuspended in a final volume of 500 µL binding buffer 1X, and propidium iodide was added to a final concentration of 5 µg/mL (Merck, Madrid, Spain). Apoptosis analyses were performed at the CCiT-UB in a Gallios flow cytometer (Beckman Coulter, Inc., Spain).

## 3. Results and Discussion

### 3.1. Synthesis of TROPY

The synthesis of TROPY was performed as described in Materials and Methods. The tricationic compound TROPY was obtained in an 80% yield by a nucleophilic substitution reaction of commercially available 1,3,5-tris(bromomethyl)benzene using three equivalents of synthesized 4-oleyloxypyridine [31] (Figure 2). It should be noted that, after 48 h of reaction, single substitution products or double substitution products were never observed. The structure of TROPY was unequivocally confirmed by 1H NMR, 13C NMR, and HRMS (Appendix A).

### 3.2. Characterization of TROPY/PPRH Complexes

The capacity of TROPY to bind PPRHs was evaluated by gel retardation assays. The PPRH used carried a fluorescent label (FAM-HpScr9). Binding reactions contained a fixed quantity of PPRH and increasing amounts of TROPY. The first lane showed a band corresponding to the fluorescent intensity given by free FAM-HpScr9, which decreased progressively as the amount of TROPY in the reaction increased (Figure 3A). The size of the resulting complex was analyzed by DLS and it exhibited a hydrodynamic diameter of 144 nm (±1.87 SE) and polydispersion of 0.28 (± 0.0092SE) (Figure 3B). Additionally, the charge of TROPY and the complex formed of TROPY–PPRH were +5.79 mV (±0.19 SE) and −8.50 mV (±0.18 SE), respectively, as determined by Doppler microelectrophoresis (Figure 3C,D).

TROPY is a tricationic compound that can form electrostatic associations with DNA, as demonstrated by gel retardation assays. The formation of these complexes protects the DNA from degradation by nucleases and confers resistance against the different physicochemical conditions of the environment [16], facilitating the stability and delivery of the PPRH. TROPY is an amphipathic compound formed by a pyridinium polar part [37,38] with a trioleyl chain. Because of this oleyloxypyridinium moiety, TROPY can help cross the phospholipid bilayer of the cell membrane. Since the size of the nanoparticles is under 200 nm, the TROPY–DNA complex may cross the membrane by an endocytosis mechanism.

The cationic nature of both compounds DOPY and TROPY provided by the pyridinium rings electrostatically establishes the interaction with the polyanionic PPRH molecules. Structurally, the two compounds differ in that DOPY has two positive charges while TROPY has three. In addition, TROPY contains an extra oleyl moiety that can increase cellular uptake.

### 3.3. Internalization of PPRH into Mammalian Cells by Fluorescence and Confocal Microscopy

To monitor the entrance of TROPY–PPRH complexes inside the cells, incubations were performed using a PPRH labeled with green fluorescence (FAM–HpScr-9) plus TROPY. Images were taken with a fluorescence microscope (ZOE Fluorescent Cell Imager Bio Laboratories) 24 h after incubation in different cancer cell lines: PC-3 from the prostate, SH-SY5Y from neuroblastoma, and 453-MDA-MB from the breast (Figure 4A). Green fluorescence was detected in all cell lines. Cell viability was not affected by these incubations since the PPRH was an unspecific scrambled sequence. The entrance was also visualized in PC-3 cells using confocal microscopy for 24 h after incubating the cells with FAM-HpScr9 and TROPY. Green fluorescence corresponds to the nanoparticles formed by TROPY–PPRH. The confocal capture shows the entrance of the complex inside the cytoplasm and into the nucleus of PC-3 cells (Figure 4B).

The entry of the complex into cells from prostate cancer (PC-3), breast cancer (MDA-MB-453-WT), and neuroblastoma (SH-SY5Y) was detected by fluorescence microscopy at 24 h. In the case of the neuroblastoma cells (SH-SY5Y), the fluorescence was less intense, which could be attributed to the fact that these cells are hard to transfect. To corroborate the intracellular entrance, confocal microscopy was employed in PC-3 cells, where it was possible to observe the entrance of the complex inside the cells and the location of the fluorescent PPRH in the nucleus.

### 3.4. Cellular Uptake by Flow Cytometry

After observing the internalization by fluorescence microscopy, cellular uptake of FAM–HpScr-9 incubated with TROPY was evaluated by flow cytometry in PC-3, SH-SY5Y, MDA-MB-453, HepG2, SKBR-3, and VERO-E6 cells. Figure 5 shows the uptake values of the PPRH upon incubation with TROPY at a defined concentration for each cell line in comparison to the control cells (CNT).

Different cell lines were assayed, measuring the percentage of positively transfected cells and the X-Mean intensity value 24 h after transfection (Table 2).

The internalization of the complex in all cell lines tested was assessed by flow cytometry. The percentage of transfected cells was higher when using TROPY compared to DOTAP for all of the cell lines. SH-SY5Y exhibited a lower percentage of positive cells and X-Mean since they are cells that are hard to transfect. This result agrees with the images taken by fluorescence microscopy. On the other hand, PC-3 and VERO-E6 presented the best levels of internalization of the complex, with a fluorescence intensity higher than the other cell lines. The PC-3 cell line was selected for further analysis since it showed the highest percentage of human transfected cells using TROPY.

We also analyzed the mechanisms involved in the internalization of TROPY/PPRH complexes by transfecting PC3 cells with FAM-HpScr9 either in the presence or the absence of different endocytic pathway inhibitors (Figure 5B). After 3.5 h of transfection, the TROPY/PPRH complex internalization was significantly reduced with either the clathrin-dependent endocytosis inhibitor Dynasore (75 μM) or the caveolin-mediated endocytosis inhibitor Genistein (185 μM), in PC-3 cells. The treatment with Dynasore produced a decrease in the PC-3 fluorescent cells of 84% relative to untreated cells (Figure 5B). Furthermore, the fluorescence mean values were reduced by 93% in PC-3 cells relative to untreated cells (Figure 5B). In the case of the Genistein treatment, the PC-3 fluorescent cells decreased by 55% (Figure 5B) and the fluorescence mean values were reduced by 85% in PC-3 cells (Figure 5B). No significant decrease was observed in either the percentage of fluorescent cells or the fluorescent mean in PC3 cells treated with the macropinocytosis inhibitor EIPA (33 μM) (Figure 5B).

### 3.5. Effect on Cell Viability and Survivin Expression Caused by PPRH against Survivin Transfected Using TROPY

Once we had demonstrated the internalization of the complex, we studied the effect on cell viability caused by the incubation of a specific PPRH (HpsPr-C) with TROPY. Previous studies in our group demonstrated that gene silencing of survivin using HpsPr-C resulted in a decrease in prostate cancer cell PC-3 survival both in vitro and in vivo when transfected with DOTAP. Therefore, we used this cell line to compare the effect of transfecting HpsPr-C with TROPY. First, the absence of toxicity resulting from the transfection of TROPY alone was determined. Then, we quantified the decrease in cell viability caused by the incubation with PPRH HpsPr-C using TROPY (Figure 6).

The concentrations of TROPY used in combination with the PPRH were 1 µg/mL and 2 µg/mL since they were not toxic on their own and they did not cause a significant decrease in viability. Prostate cancer cells incubated with the PPRH using either 1 µg/mL or 2 µg/mL of TROPY, showed a significant decrease in viability of 53% and 81%, respectively. The effect of HpsPr-C transfected with 2 µg/mL (1.4 µM) of TROPY was similar to that caused by 10 µM DOTAP, which corresponds to 7 times less compared to the commercial transfection agent [15]. The effect obtained with TROPY was moderately better than with DOPY, which needed a concentration of 2.1 µM to attain the same effect [31].

Additionally, we analyzed the survivin expression upon transfection with the TROPY–PPRH complexes at both the mRNA and protein levels. As can be seen in Figure 6B, the transfection of HpsPr-C decreased survivin mRNA levels to 61% at 24 h and 50% at 48 h after transfection, respectively. The reduction in survivin mRNA levels was translated into a reduction of 78% of the corresponding protein levels (Figure 6C).

### 3.6. Effect on Apoptosis Caused by PPRH against Survivin Transfected with TROPY

Apoptosis was determined by flow cytometry in PC-3 cells incubated with annexin V 24 h after transfection. The PPRH used was HpsPr-C against the survivin antiapoptotic gene and using TROPY as the transfectant agent (Figure 7).

The PPRH used throughout the project has been HpsPr-C targeting the survivin gene, which is overexpressed in many types of cancer such as breast [39], gastric [40,41], prostate [42]), neuroblastoma [43,44] and osteosarcoma [45]. Since survivin is an antiapoptotic protein, gene silencing with HpsPr-C decreases its expression, resulting in an increase in apoptosis and a decrease in the viability of tumor cells [15]. In our conditions using TROPY, an increase in apoptosis of 46% and 65% was observed after transfection of the PPRH with 2 µg/mL and 5 µg/mL of the transfection agent, respectively. This confirmed the delivery of the hairpin to the nucleus exerted by TROPY and the effectiveness of the complex to decrease cell viability. A comparison of the levels of apoptosis in PC-3 cells transfected with HpsPr-C either using 7.75 µg/mL of DOTAP (10 µM) or 1 µg/mL of TROPY (0.7 µM) was also performed. In these conditions, apoptosis increased to 3.1% with DOTAP, whereas it reached 10% in the case of TROPY, confirming a more efficient silencing of the antiapoptotic target gene by HpsPr-C with 14 times less transfection agent.

### 3.7. Transfection of Plasmid DNA into Mammalian Cells

To explore the capability of TROPY to also transfect plasmid DNA, we used the pDCH1P construction [46], a dihydrofolate reductase (dhfr) minigene cloned into pUC119 [47], transfected into a dhfr-negative CHO cell line (DG44). A dose-response titration was performed using pDCH1P plus increasing amounts of TROPY in 35 mm diameter dishes. After 30 h, which included the expression time of the plasmid, the culture medium was changed into a DHFR metabolic selective medium (RPMI plus dialyzed serum) lacking the final products of DHFR activity, i.e., hypoxanthine and thymidine. The culture medium was changed every 5 days until the appearance of colonies. The results are shown in Figure 8 where we can observe the increasing number of DHFR-positive colonies depending on the amount of TROPY used in the plasmid transfection. Additionally, transfections were also carried out in 10 cm dishes with 500,000 starting cells transfected with a concentration of 200 µg/mL of pDCH1P using TROPY at 3 µg/mL. After 30 h of transfection plus expression, the medium was changed to DHFR selective medium until the formation of viable colonies, which were then fixed with formaldehyde, stained with crystal violet, and counted. In this instance, the transfection efficiency was 420 colonies/µg of the plasmid.

## 4. Conclusions

In this work, we describe the synthesis and evaluation of the efficacy of a novel oleyloxypyridinium moiety (TROPY) as a delivery agent for therapeutic oligonucleotides in mammalian cells. TROPY, with three positive charges provided by the pyridinium ring, forms electrostatic complexes with the negative charges of the phosphodiester bonds in the DNA (PPRH). TROPY–PPRH nanoparticles present a hydrodynamic diameter of 144 nm, as determined by DLS, with low size distribution. The complex TROPY–PPRH is internalized in cancer cells from prostate PC-3, breast MDA-MB-453 and SKBR-3, neuroblastoma SH-SY5Y, liver HepG2, and in kidney epithelial cells from African green monkey VERO-E6 through clathrin and caveolin-mediated endocytosis. The uptake was corroborated by fluorescence microscopy, flow cytometry, and confocal microscopy analyses using a PPRH labeled with green fluorescence. Likewise, confocal captures show the entrance of the complex both in the cytoplasm and the nucleus. TROPY can efficiently transfect therapeutic oligonucleotides at a concentration of 1.4 µM, 7 times lower than the commercial transfection agent DOTAP. A validated PPRH designed against the antiapoptotic gene *survivin* was used as a model of a therapeutic oligonucleotide. The complex formed by TROPY–HpsPr-C decreases cell viability, produces a high increase in apoptosis, and reduces the levels of mRNA and protein of the target gene *survivin*. TROPY is also able to lipofect plasmid DNA into mammalian cells allowing the stable transfection of a minigene with high efficiency, as demonstrated by the generation of viable colonies upon lipofection of a *dhfr* plasmid into *dhfr*-negative cells, cultured in metabolic selective medium. We conclude that TROPY is an efficient transfecting agent that allows the delivery of therapeutic oligonucleotides, such as PPRHs and plasmid DNA, inside mammalian cells. Further work will be needed to evaluate the effectiveness of TROPY in in vivo approaches to explore its potential in the clinical application of oligonucleotides as therapeutic tools.

## Figures and Tables

**Figure 1 pharmaceutics-15-00420-f001:**
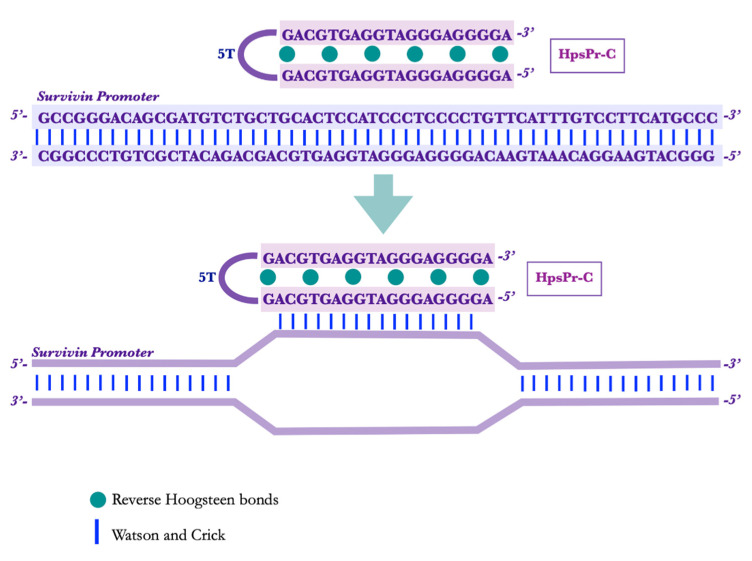
HpsPr-C scheme of action. The PPRH directed against the promoter of the *survivin* gene (HpsPr-C) binds to a polypyrimidine stretch found in the coding strand of the gene, forming a triplex structure and displacing the reverse strand in that region, resulting in an inhibition of the transcription process.

**Figure 2 pharmaceutics-15-00420-f002:**
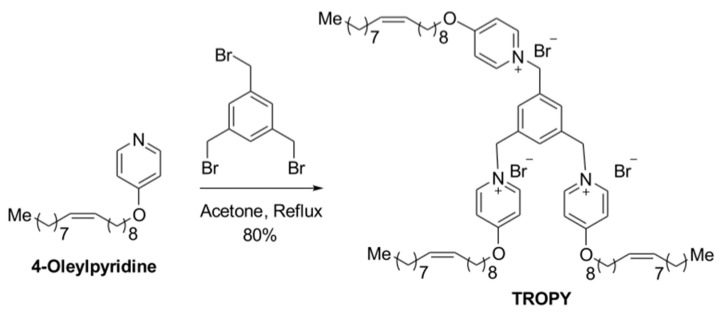
Synthesis of TROPY. Tricationic compound was obtained from commercially available 1,3,5-tris(bromomethyl)benzene and three equivalents of 4-oleylpyridine [31].

**Figure 3 pharmaceutics-15-00420-f003:**
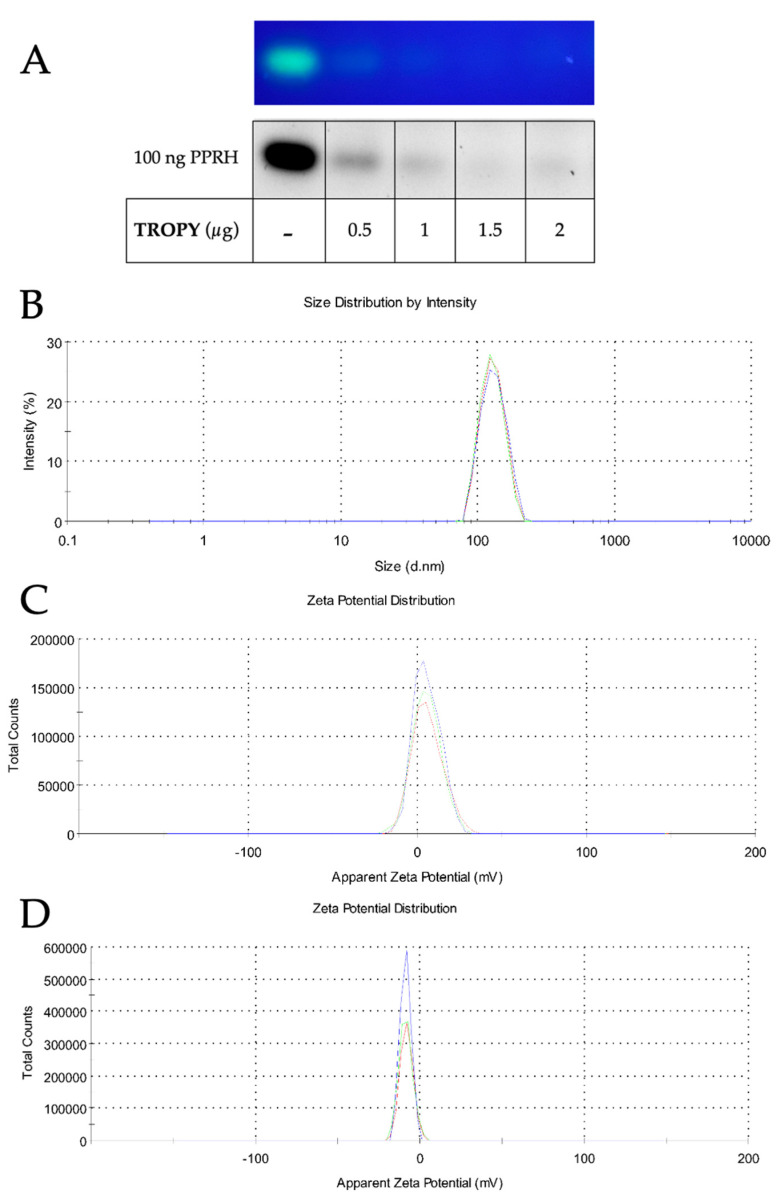
Characterization of the complex formed between PPRH and TROPY. (**A**) The binding capacity of TROPY was determined using 100 ng of a FAM-labeled PPRH incubated with the indicated amounts of the transfection agent. The mixture was carried out in PBS 1X for 20 min in a final volume of 10 µL, to which was then added 2 µL of 6× sample buffer, and the mixture was resolved electrophoretically in 0.8% agarose gels. Bands were visualized under UV light in a Bio-Rad Gel Doc EZ^TM^ Imager. The bands are shown both as fluorescent and in black and white. The image shown is representative of three electrophoreses. (**B**) The size of the complex was determined by DLS using 1.25 µg/mL of TROPY incubated with 100 nM PPRH. (**C**) The charge of TROPY alone and (**D**) TROPY–PPRH complexes were determined by Doppler microelectrophoresis using 1.5 µg/mL of TROPY with 100 nM PPRH. Determinations are the mean ± SE of three experiments. The 3 peaks with colours represent each one of the 3 measurements of the triplicate determination.

**Figure 4 pharmaceutics-15-00420-f004:**
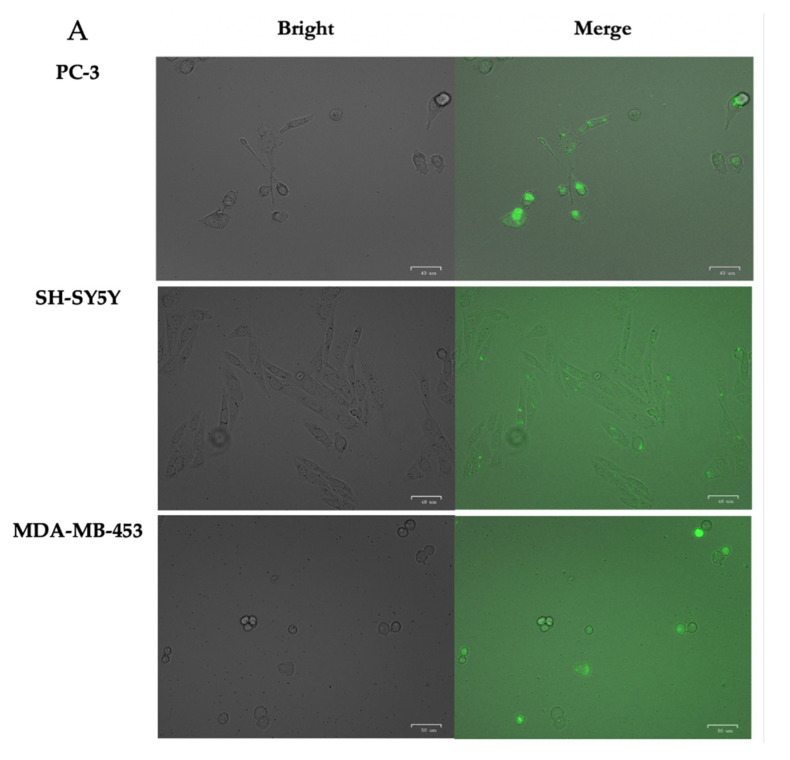
Cell Internalization of PPRHs. (**A**) Images from bright and merge fluorescence microscopy were taken from the three cell lines indicated upon incubation for 24 h with 100 nM FAM-HpScr9 plus 1.5 µg of TROPY. (**B**) Confocal microscopy after incubating PC3 cells with 100 nM FAM-HpScr9 plus 1.5 µg of TROPY. Cells were incubated with WGA 555 to identify cell membranes in red and with Hoechst to label nuclei in blue (upper image). Green complexes of PPRH–TROPY were visualized within the cytoplasm and in the nucleus (lower image).

**Figure 5 pharmaceutics-15-00420-f005:**
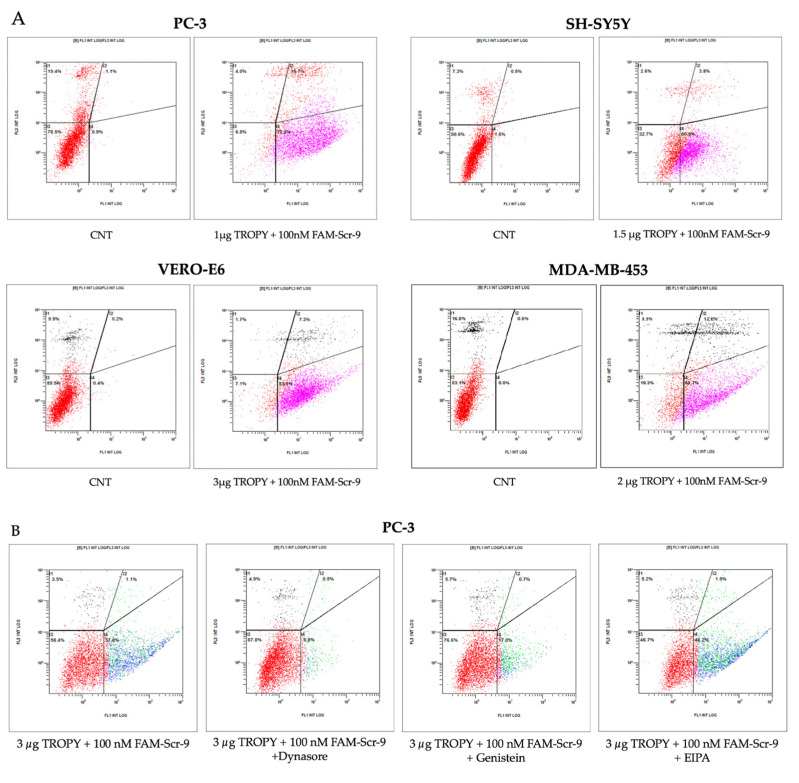
Cellular uptake by flow cytometry. (**A**) The uptake of the fluorescent PPRH is shown in four different cell lines incubated in 1 mL of culture medium for 24h with the indicated amounts of TROPY. The screening generated by flow cytometry analyses divided the cell population into quadrants. The upper and lower right quadrants represented the internalization of positive-gated cells with the fluorescent TROPY–HpScr9 complex. The lower left represents the autofluorescence of the cells, and the upper left, the dead cells stained with propidium iodide. In all cases, we visualized a shift of the cell population to the right when comparing the fluorescence of the incubated cells to that of the control. The increment was quantified by the green fluorescence of FAM-HpScr9. (**B**) Endocytic pathways involved in the internalization of the TROPY/PPRH complexes. PC-3 cells were incubated for 1 h with 75 μM of Dynasore, 185 μM Genistein, or 33 μM EIPA and subsequently transfected with 100 nM of the FAM-HpsPr-C PPRH using 2.1 μM of TROPY. After 3.5 h of incubation, the percentage of fluorescent cells and the mean fluorescence of the PC3 cells were determined by flow cytometry. In (**A**), the fucsia represents the internalized fluorescent cells, whereas in (**B**), the blue colour represents the internalized fluorescent cells in the experiments using endocytic pathways inhibitors. The red in CNT is the control.

**Figure 6 pharmaceutics-15-00420-f006:**
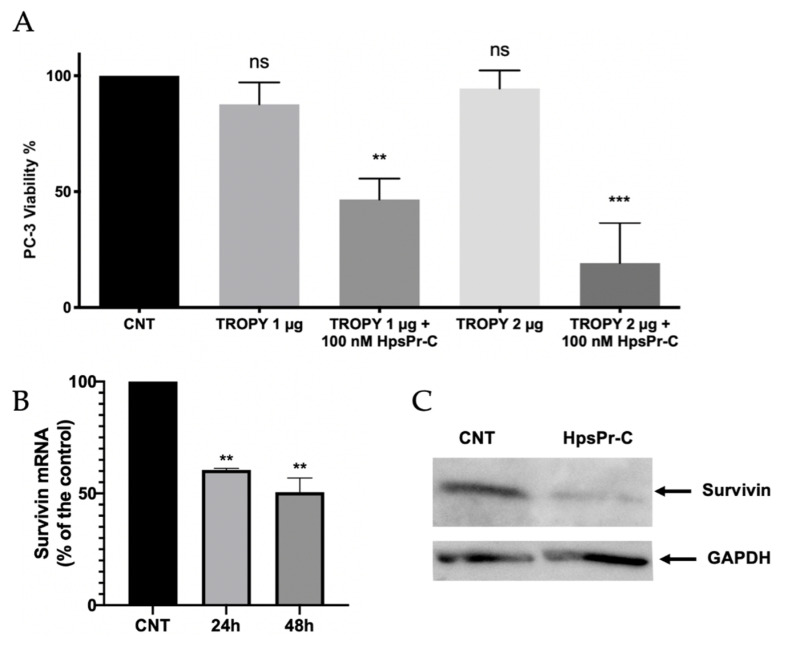
(**A**) The effect caused by HpsPr-C transfected using the indicated amounts of TROPY on prostate cancer cell viability. The intrinsic toxicity of TROPY was evaluated at 1 µg/mL and 2 µg/mL of the transfectant agent. Two different concentrations of TROPY were used to explore the effect of 100 nM HpsPr-C. Experiments were performed in triplicate. Statistical significance was calculated using one-way ANOVA with multiple comparisons test (ns, non significant, ** *p* < 0.01, *** *p* < 0.001). (**B**) Survivin mRNA levels were determined by RT-qPCR from PC-3 control cells or transfected with 100 nM HpsPr-C and 2 µg/mL of TROPY for 24 and 48 h. Experiments were performed in duplicate. Statistical significance was calculated using one-way ANOVA with multiple comparisons test (** *p* < 0.01). (**C**) Representative blot of the signal corresponding to Survivin protein levels in control cells and HpsPr-C-transfected for 24 h as determined by Western blot; GAPDH was used for the normalization of the results. Experiments were performed in duplicate.

**Figure 7 pharmaceutics-15-00420-f007:**
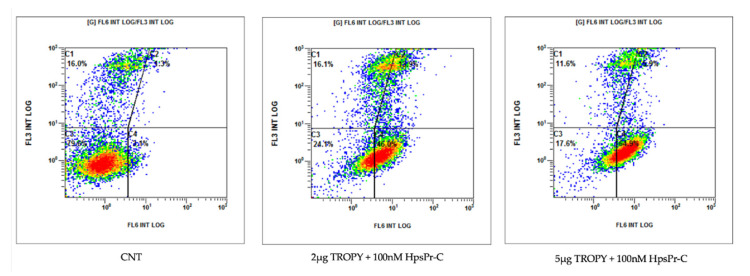
Apoptosis levels in PC-3 cells transfected with HpsPr-C using TROPY. Figures show the population of apoptotic cells upon transfection of 100 nM of PPRH with increasing amounts of TROPY (2 µg/mL and 5 µg/mL). The displacement of the population of apoptotic cells is shown in the right lower quadrant. Images are representative of three experiments.

**Figure 8 pharmaceutics-15-00420-f008:**
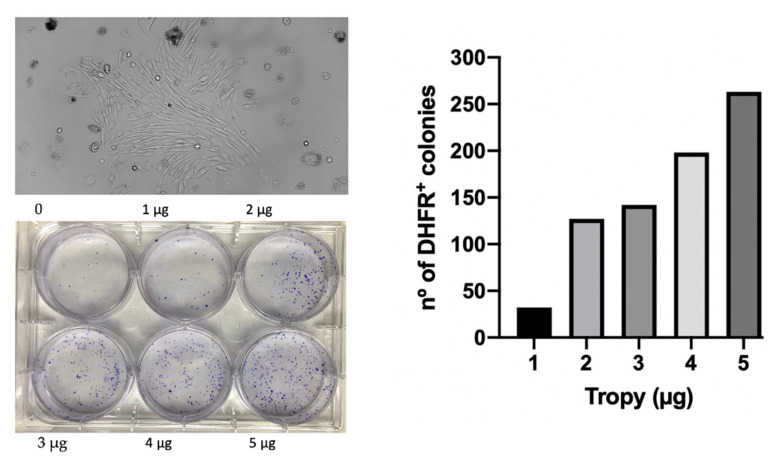
The efficiency of plasmid transfection using TROPY. The pDCH1P plasmid (300 ng) was transfected with the indicated amounts of TROPY in 1 mL of full F12 medium in 35 mm diameter dishes. After 30 h, the medium was replaced with DHFR selective medium lacking hypoxanthine and thymidine. A microscope view of a representative colony after metabolic selection is shown. Colonies were stained with crystal violet and counted.

**Table 1 pharmaceutics-15-00420-t001:** Oligonucleotides used in this study.

Gene	PPRH	Sequence 5′–3′	Location
Survivin	HpsPr-C	AGGGGAGGGATGGAGTGCAG T T T AGGGGAGGGATGGAGTGCAG T T	Promoter
Survivin	FAM-HpsPr-C	[6FAM]AGGGGAGGGATGGAGTGCAG T T T AGGGGAGGGATGGAGTGCAG T T	Promoter
Scr-9	FAM-HpScr9	AAGAAGAAGAAGAGAAGAA T T T AAGAAGAAGAAGAGAAGAA T T	-

**Table 2 pharmaceutics-15-00420-t002:** Cell lines transfected with TROPY or DOTAP.

Cell Line	TROPY			DOTAP 10 µM	
	µg	Transfected Cells (%)	X-Mean	Transfected Cells (%)	X-Mean
PC-3	1.0	89.2	40.2	69.4	64.4
SH-SY5Y	1.5	64.7	8.76	n.d	n.d
MDA-MB-453	2.0	71.1	33.8	n.d	n.d
HepG2	2.0	37.1	70.8	7.6	33.0
SKBR-3	2.0	86.1	63.5	47.6	35.0
VERO-E6	3.0	91.1	62.2	69.2	9.3

Cell lines assayed with the indicated amounts of TROPY or DOTAP 10 µM. For each cell line, we adjusted the concentration of TROPY incubated with 100 nM FAM-HpScr9. The percentage of positive cells corresponded to cells incubated with the complex and labeled with green fluorescence internalized, as determined by flow cytometry.

## Data Availability

All data are presented within the submitted manuscript.

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
