# Peer review of "Trioleyl Pyridinium, a Cationic Transfection Agent for the Lipofection of Therapeutic Oligonucleotides into Mammalian Cells"

_pharmaceutics, 2023, doi:10.3390/pharmaceutics15020420_

Round 1

Reviewer 1 Report

This is a well-written study about trioleyl pyridinium for the lipofection of therapeutic oligonucleotides in mammalian cells. I recommend it for publication after the following minor points are addressed.

1. Line 66, 'chemically synthesized molecules', several studies (Biomaterials 35 (27), 7909-7918; Acta biomaterialia 41 (2016): 293-301) should be included to support the claim. 

2. 2.2. Preparation of TROPY. Single substitution and double substitution products of 1,3,5-Tris(bromomethyl)benzene may have already precipitated from the solvent, therefore triple substitution product is not easy to be obtained. Could the author discuss about it?

3. The resolution of figures 3,5,7 should be improved to a higher level.

4. Why the zeta potential of the nanoparticles are negative? Are there enough positively charged materials to condense the DNA?

Author Response

Comments and Suggestions for Authors

This is a well-written study about trioleyl pyridinium for the lipofection of therapeutic oligonucleotides in mammalian cells. I recommend it for publication after the following minor points are addressed.

1.Line 66, 'chemically synthesized molecules', several studies (Biomaterials 35 (27), 7909-7918; Acta biomaterialia 41 (2016): 293-301) should be included to support the claim. 

We have revised the sentence of line 66 “chemically synthesized molecules” and the reference proposed by the reviewer have been included to support the claim.

Additionally, we have also added two other references: the review “Belmadi, N. et al. Synthetic vectors for gene delivery: An overwiew of their evolution depending on routes of administration. Biotechnol. J. 2015, 10, 1370-1389, doi: 10.1002/biot.20140084”1 and the article “Kim, J et al. Synthesis and application of poly(ethylene glycol)-co-poly(b-aminoester)copolymers for small cell lung cancer gene therapy. Acta Biomaterialia 2016, 41, 293-301, doi: 10.1016/j.actbio.2016.05.040”.

  1. Preparation of TROPY. Single substitution and double substitution products of 1,3,5-Tris(bromomethyl)benzene may have already precipitated from the solvent, therefore triple substitution product is not easy to be obtained. Could the author discuss about it?

Although it is true that at short reaction times the formation of more than one product is observed (possibly a mixture of single, double, and triple substitution products), after 48h the only product observed by 1H-NMR was the triple substitution product.

The integration of the signal at 5.71 ppm (brs, 6H) unequivocally confirms the entry of three chains. The non-observation of any signal at 4.45 ppm (PhCH2Br) excludes the formation of the mono- or disubstituted products. Also, the chemical shift, the multiplicity, and the integration of the signals at 5.36-5.33 (m, 6H, CH=CH) and 4.19 (t, 6H, CH2O) unequivocally corroborate the formation of TROPY. In addition, HRMS indicates only the presence of the three positive charges [MW (C78H126N3O3)3+ 1152.9783 g/mol, HMRS (ESI-TOF) m/z Calcd 1152/3= 384.3261; Found 384.3266].

To clarify this result, we have added a new paragraph describing the reaction in section 3.1 Synthesis of TROPY, and the text of Figure 2 has been simplified.

  1. The resolution of figures 3,5,7 should be improved to a higher level.

As suggested by the referee and the editor, we have uploaded all the figures as .TIFF files with high resolution

  1. Why the zeta potential of the nanoparticles are negative? Are there enough positively charged materials to condense the DNA?

The zeta potential of the transfection agent TROPY alone was indeed positive. However, when it was complexed with a final concentration of 100 nM, the zeta potential of the nanoparticles was negative.

This information has been added to the results section of the manuscript including the graph of zeta potential of TROPY alone as Figure 3C.

Reviewer 2 Report

In this study the authors described a novel method based on TROPY (Trioleyl pyridinium, a cationic transfection agent) to bind and deliver nucleic acids (DNA and oligonucleotides) with therapeutic potentials in mammalian cells. They observed that this nanocomplex was able to transfect an oligonucleotide against the antiapoptotic survivin gene and consequently they reported changes in cell viability and apoptosis, speculating that this phenotype was due to the antisense effect of such oligo. In addition, they reported that the nanocomplex was also able to transfect plasmid DNA as demonstrated by colony formation assay. Overall the study is very interesting, and provides a novel tool to deliver nucleic acids in mammalian cells, that is a limitant issue to fully exploit their therapeutic potential. However, there are some weaknesses that should be considered and resolved before publication. In particular, this reviewer consider that the authors should -increase literature analysis relative to other similar delivery methods and compare the effectiveness of their methods with the other ones; -use additional control, in particular compare the effectiveness of commercial delivery methods cationic based with TROPY; -investigate the mechanisms by which TROPY complex enter in cell lines; and -analyze the gene expression of surviv after transfection (this is very important); -deepening the possibility of clinical and market translation of this nano-product.

The specific points are reported here:

1) The authors should mention in the introduction that other nanotherapeutic systems have been recently developed for nucleic acid delivery in mammalian cells. 10.1038/s41576-021-00439-4, 10.1038/s41467-017-01386-7, 10.3390/pharmaceutics13122067. It would be interesting trying to compare the systems used by the authors with those already employed, describing the advantages and disadvantages of each one (for example, why is it better TROPY complex with respect to metal-based nanoparticles to deliver nucleic acids?).

2) It would be very interesting that authors show the differences between TROPY with another liposomal-cationic system currently available in the market to deliver nucleic acids (like DOTAP or lipofectamine). For example, a comparison of the transfection efficiency of lipofectamine/DOTAP versus TROPY in several cell lines should be provided. I recommend to the authors to add lipofectamine or DOTAP as control in most of the experiments carried out (surely in fluorescence microscopy and viability/apoptosis assay). A detailed comparison of such results and the discussion regarding the differences between TROPY with other liposomal systems would increase the value of the study.

3) The authors showed that the TROPY-PPRH complex can be internalized in the cells through experiments of fluorescence microscopy and viability studies to evaluate the effect of oligonucleotides. Effectively these are promising results, but more experiments should be required to confirm the effectiveness of the complex: 

3.1) The authors should evaluate by western blot (or qPCR) that survivin gene is effectively modulated, this would allow them to prove that the results observed are effectively due to change in gene expression (that I guess to be the master goal of the study). 

3.2) In addition, they should investigate the mechanisms by which TROPY complex enter in the cells by treating them with inhibitors of clathrin-mediated and/or caveolin mediated endocytosis (like chlorpromazine - 10.1083/jcb.123.5.1107 - and filipin - 10.1083/jcb.141.4.905 ). If the authors can not provide these experimental results, they should justify why not, and to discuss deeply and carefully their hypothesis and speculation about the mechanism of internalization, also comparing other literature and their previous studies.

4) Have a mention about the possibility that the results deriving from this study might be applied in animal models, and speculate about its clinical & market potential to make the study more appealing.

Author Response

Comments and Suggestions for Authors

In this study the authors described a novel method based on TROPY (Trioleyl pyridinium, a cationic transfection agent) to bind and deliver nucleic acids (DNA and oligonucleotides) with therapeutic potentials in mammalian cells. They observed that this nanocomplex was able to transfect an oligonucleotide against the antiapoptotic survivin gene and consequently they reported changes in cell viability and apoptosis, speculating that this phenotype was due to the antisense effect of such oligo. In addition, they reported that the nanocomplex was also able to transfect plasmid DNA as demonstrated by colony formation assay. Overall the study is very interesting, and provides a novel tool to deliver nucleic acids in mammalian cells, that is a limitant issue to fully exploit their therapeutic potential. However, there are some weaknesses that should be considered and resolved before publication. In particular, this reviewer consider that the authors should -increase literature analysis relative to other similar delivery methods and compare the effectiveness of their methods with the other ones; -use additional control, in particular compare the effectiveness of commercial delivery methods cationic based with TROPY; -investigate the mechanisms by which TROPY complex enter in cell lines; and -analyze the gene expression of surviv after transfection (this is very important); -deepening the possibility of clinical and market translation of this nano-product.

The specific points are reported here:

  • The authors should mention in the introduction that other nanotherapeutic systems have been recently developed for nucleic acid delivery in mammalian cells. 10.1038/s41576-021-00439-4, 10.1038/s41467-017-01386-7, 10.3390/pharmaceutics13122067. It would be interesting trying to compare the systems used by the authors with those already employed, describing the advantages and disadvantages of each one (for example, why is it better TROPY complex with respect to metal-based nanoparticles to deliver nucleic acids?).

We have added in the introduction a review and 2 articles describing recent developments of nanotherapeutic systems for nucleic acid delivery in mammalian cells. In addition, in the Results and Discussion section a comparison between TROPY and a commercial system already employed DOTAP has been included at the levels of cytotoxicity, PPRH uptake and apoptosis.

Since the aim of this work is the characterization of a new transfection agent, we feel that its comparison to other available nanotherapeutic systems other than DOTAP, our previously liposome of reference, is out of the scope of the present manuscript.

  • It would be very interesting that authors show the differences between TROPY with another liposomal-cationic system currently available in the market to deliver nucleic acids (like DOTAP or lipofectamine). For example, a comparison of the transfection efficiency of lipofectamine/DOTAP versus TROPY in several cell lines should be provided. I recommend to the authors to add lipofectamine or DOTAP as control in most of the experiments carried out (surely in fluorescence microscopy and viability/apoptosis assay). A detailed comparison of such results and the discussion regarding the differences between TROPY with other liposomal systems would increase the value of the study.

As already mentioned in our answer to Point 1 raised by the referee, the effects of TROPY compared to DOTAP, a commercial system already employed, have been included at the levels of PPRH uptake and levels of apoptosis.

Regarding the uptake of a fluorescent PPRH, a table summarizing the % of transfected cells and X-mean when using DOTAP as transfection agent for PC-3, HepG2, SKBR-3 and VERO-6 cells has been included in the new version of the manuscript for comparison.

We had already indicated that 1.4 µM of TROPY caused the same cytotoxic effect as 10 µM of DOTAP when complexed to 100 nM of PPRH.

Finally, a comparison of the levels of apoptosis in PC-3 cells transfected with HpsPr-C either using DOTAP 7.75 µg (10 µM) or TROPY 1 µg (0.7 µM) was also performed, confirming a more efficient silencing of the antiapoptotic target gene by TROPY-HpsPr-C, with 14 times less of transfection agent.

3) The authors showed that the TROPY-PPRH complex can be internalized in the cells through experiments of fluorescence microscopy and viability studies to evaluate the effect of oligonucleotides. Effectively these are promising results, but more experiments should be required to confirm the effectiveness of the complex: 

3.1) The authors should evaluate by western blot (or qPCR) that survivin gene is effectively modulated, this would allow them to prove that the results observed are effectively due to change in gene expression (that I guess to be the master goal of the study). 

As suggested by the referee, the expression of survivin upon transfection of the TROPY-PPRH complex have been determined both at the mRNA and protein levels. We show that the mRNA levels decreased by 40% at 24h after transfection and by 50% at 48h. In addition, the protein levels are decreased by 78% after 24 h of transfection. These results have been added to the new version of the manuscript.

3.2) In addition, they should investigate the mechanisms by which TROPY complex enter in the cells by treating them with inhibitors of clathrin-mediated and/or caveolin mediated endocytosis (like chlorpromazine - 10.1083/jcb.123.5.1107 - and filipin - 10.1083/jcb.141.4.905 ). If the authors can not provide these experimental results, they should justify why not, and to discuss deeply and carefully their hypothesis and speculation about the mechanism of internalization, also comparing other literature and their previous studies.

To evaluate whether the clathrin-mediated and/or caveolin mediated endocytosis were responsible for the entrance of the TROPY complexes into the cells, we incubated the cells for 1 hour with the following inhibitors: 75 μM of the clathrin-dependent endocytosis

inhibitor Dynasore, 185 μM of the caveolin-mediated endocytosis inhibitor Genistein, or 33 μM of the micropinocytosis inhibitor 5-(N-ethyl-N-isopropyl) amiloride (EIPA) and analyze the uptake of a fluoresencent PPRH by flow cytometry after 3.5h of transfection. In these conditions, the uptake of the PPRH was reduced by both Dynasore and Genistein but remained unaffected by EIPA, indicating that both clathrin and caveolin mediated endocytosis take part in the entrance of the TROPY complexes.

These results have been added to the new version of the manuscript

4) Have a mention about the possibility that the results deriving from this study might be applied in animal models, and speculate about its clinical & market potential to make the study more appealing.

The possibility of using the transfection agent TROPY in in vivo settings has been mentioned in the Conclusion as well as its potential in the clinical application of oligonucleotides as therapeutic tools.

Round 2

Reviewer 2 Report

The authors performed mostly of the experiments required by this reviewer and provided interesting data that strongly support their conclusions. They included analysis of survivin gene with TROPY-PPRH complex (mRNA and WB), compared the transfection efficiency with DOTAP as commercial transfection agent, and they analyzed the mechanism of internalization of their nanosystem by using different inhibitors. The manuscript is definitively improved.